# Evaluating the Impact of Concrete Design on the Effectiveness of the Electrochemical Chloride Extraction Process

**DOI:** 10.3390/ma16020666

**Published:** 2023-01-10

**Authors:** Zofia Szweda

**Affiliations:** Department of Building Structures, Faculty of Civil Engineering, Silesian University of Technology, 44-100 Gliwice, Poland; zofia.szweda@polsl.pl

**Keywords:** chloride migration, corrosion risk, chloride extraction time, electrochemical chloride extraction, reinforced concrete structures, repair, rehabilitation

## Abstract

This paper presents a simple comparative method for evaluating the impact of concrete design on the effectiveness of repair with the electrochemical chloride extraction (ECE) process of reinforced concrete structures. This comparison covered two concretes with different types of used cement. Penetration of chloride ions to induce corrosion processes was accelerated with the electric field. However, the corrosion process itself occurred naturally. When the corrosion process was found to pose a risk to the reinforcement, the profile of chloride ion concentration was determined at the depth of concrete cover. Corrosion current intensity during migration and extraction processes of chloride ions was measured with the LPR method. Then, this serious condition for the structure was repaired with electrochemical chloride extraction. Rates of chloride extraction were determined from the derived concentration profiles. It should be noted that the critical concentration C_crit_ = 0.4% at the rebar surface was reached after 21 days of the migration process. Moreover, after the same time of extraction, the concentration was reduced by 95% at the rebar surface, which could suggest that extraction rate was slower than chloride ion migration to concrete within the electric field. Using the migration coefficient for predicting the extraction time, as well as ignoring the variability of the extraction coefficient and the initial concentration over time, may result in too short or unnecessarily long extraction times.

## 1. Introduction

The presence of chloride ions in de-icing agents and the coastal environment, as well as groundwater and municipal wastewater, is defined as the main cause of the risk of corrosion of reinforced concrete structures. Exceeding the critical concentration of chlorides at the depth of reinforcement leads to the initiation and development of a very dangerous type of steel corrosion, the so-called pitting corrosion. Chloride ions contribute to the destruction of the protective layer of passive reinforcing steel in concrete. Then, the mechanism of destruction consists of reducing the cross-section of the reinforcing bars and the accumulation of corrosion products at the surface of steel. The reduced cross-section leads to structural failure. On the other hand, the accumulated products of corrosion cause the bursting of the concrete cover. As the reinforcement is exposed to weather conditions, the corrosion processes accelerate [1].

Unfortunately, this critical content of chloride ions at the surface of the reinforcement is not clearly defined. According to the European [2] and British Standards, [3] C_crit_ = 0.4% of cement by weight in reinforced concrete structures and C_crit_ = 0.1% of cement by weight in prestressed structures are considered as the critical chloride content, at which corrosion process can develop. The American Standards [4,5] allow for C_crit_ = 0.1% of cement by weight in reinforced concrete structures and C_crit_ = 0.06% of cement by weight in prestressed structures. Moreover, the standard [6] allows for C_crit_ = 0.2% of cement by weight in reinforced concrete structures and C_crit_ = 0.08% of cement by weight in prestressed structures.

The content of hydroxide ions in the concrete pore liquid is an additional factor which affects the initiation rate of corrosion processes. Haussman [7] developed the relationship between chloride and hydroxide ions by determining critical value, at which corrosion processes can be observed, as the ratio of chloride to hydroxide ions equal to 0.6. In numerous papers [8,9,10], the relationship between the content of free chloride ions and concentration of hydroxyl ions in concrete was used to express critical content of chlorides, but its value varied within a wide range from 0.3 to 40.

Concrete additives, such as pulverised fly ash and ground granulated blast furnace slag, affected the threshold values of chloride ion concentration. Differences were also observed in determining the threshold value of chloride ion content depending on whether they were added to concrete mix or originated from the external environment [11]. 

Another difficulty is the fact that corrosion processes of the reinforcement are not visible to the naked eye because they run under the surface of concrete cover. Qualitative evaluation of reinforcement corrosion can be performed with such electrochemical methods such as measuring potential and resistivity of concrete cover [12,13,14]. 

The assessed corrosion risk to the structure requires its immediate rehabilitation [15]. Since 1985, the ECE (Electrochemical Chloride Extraction) method has been used in many countries for reinforced concrete structures contaminated with chlorides [16,17,18]. 

It is also very important to precisely determine the development of corrosion in reinforced concrete structures before and after the application of electrochemical repair method. The repair time should be predicted more precisely and its effectiveness should be evaluated while analysing corrosion measurements using the LPR and EIS techniques [19].

Duration of the extraction process has been so far based on the migration coefficient determined from the migration process of chlorides in concrete [20]. Apart from poor effectiveness and repeatability of the standard methods of determining the migration coefficient, the obtained values of this coefficient could not be used to precisely determine the duration of the extraction process [21]. In the majority of papers describing the extraction method, this process was performed on laboratory specimens of cement pastes and grouts. Some papers presented the works conducted on the specimens of concrete with the ribbed reinforcement [22], however the extraction process is usually performed on the specimens of concrete mix, into which chloride ions are added directly after being dissolved in batched water [23]. There are still very few examples of tests performed under the laboratory conditions similar to the original ones, that is, with concrete and ribbed reinforcement and associated with chloride ions migrating from the external environment. [24].

This paper presents a simple comparative method for evaluating the impact of concrete design on the effectiveness of electrochemical extraction chlorides from concrete. This method was introduced in the paper [19] for one concrete. In this paper, the method was applied to a different type of concrete and used to compare both concretes at the same time. The tests described in that paper had two aims: the first was to test the method on the second concrete (it will be necessary to test this method on many different concretes); the second aim was to compare the effectiveness of Electrochemical Chloride Extraction (ECE) in both concretes using this method. 

In this method, chloride ions were at first added from the external environment. The process of ion penetration was accelerated with the action of electric field. The specimens consisted of tested concrete types and contained reinforcement made of ribbed steel. When the corrosion process was found to pose a risk to the reinforcement, the profile of chloride ion concentration was determined at the depth of concrete cover. Then, this serious condition for the structure was repaired with electrochemical chloride extraction (ECE). Corrosion current intensity during migration and extraction processes of chloride ions were measured with the LPR method. After obtaining the satisfactory values of corrosion current, distribution of chloride ion concentration was checked at the depth of concrete cover. Then, the extraction coefficient was calculated from the distributed concentration of chloride ions at the depth of concrete cover after the relevant duration of this process. Linear variability of such values as boundary concentration of chloride ions and extraction coefficient over time were also included in this method in a simplified way. The extraction process was numerically modelled. This new approach is used to more precisely determine values of extraction coefficients for existing structures and to assess the effective duration of chloride extractions both for the existing structure and the process of designing and verifying properties of new concrete types. 

## 2. Materials

The tests were performed on two types of concrete mix. Concretes C1 and C2 were made of ordinary concrete with different type of used cement. Specimens made of the same preparations were used in the work [25] to determine values of the diffusion coefficient of chloride ions. Concrete C1 contained CEM I 42.5 R cement. While concrete C2 contained blast-furnace cement with lowered content of alkalis—CEM III/A 42.5 N-LH/HSR/NA. The detailed compositions of mixes are presented in Table 1.

## 3. Test Methods

All of the tests were conducted and the specimens were prepared at the Laboratory of Civil Engineering of the Silesian University of Technology. Six cylindrical-shaped test specimens 1 with a diameter of 100 mm and a height of 60 mm were prepared from each concrete type. Ribbed rebars 2 with ø12 mm, made of steel B500SP, were placed in these specimens in the direction perpendicular to the cylinder axis. The most common diameter used for the main reinforcement was 12 mm and the concrete cover of 20 mm was adjusted to this diameter. The specimens were prepared as described in the paper [19]. Figure 1 shows the specimens prepared for testing during curing, before attaching plastic tanks made of PVC pipes to the upper surface of these elements. 

### 3.1. Migration of Chloride Ions Accelerated with the LPR Method with Simultaneous Control over Corrosion

Prior to chloride diffusion to concrete accelerated by the electric field, the polarization tests were performed on all of the specimens with the LPR method (Figure 2) to determine corrosion potential of the specimen reinforcement in the passive state. Electrochemical measurements are greatly influenced by humidity and temperature. Therefore, all electrochemical studies were made under the same conditions. Before the tests, the specimens were immersed in water for 72 h in order to stabilize the half-cell potential and avoid overload in the potentiostat. In this case, the corrosion rate was not controlled by oxygen diffusion to the steel surface [26]. The measurements were performed in a three-electrode arrangement, where steel rebar was used as the working electrode 2. The counter electrode 4 was made of stainless-steel sheet, whose shape was adjusted to the test specimens. The reference electrode 5 (Cl^−^/AgCl,Ag) was placed on the cylinder surface. It rested against walls of the plastic tank tightly fixed to the specimen. To provide satisfactory conductivity, a felt separator soaked with distilled water was placed both on the top element of the tank and in the bottom tank. The specimens were soaked with water by immersion for ca. 72 h. Then, the LPR tests were performed using the potentiostat 6 Gamry Reference 600 by Gamry Instruments, Warminster, Pennsylvania, United States of America in the potentiostatic mode within a range of frequencies 10 mHz–100 kHz at an amplitude of 10 mV over the corrosion potential of the reinforcement.

After taking from water, the specimens were connected to the potentiostat 6 and changes in gradually stabilizing potential were observed with the reference electrode 5 for 60–120 min. When potential changes were at the level of 0.1 mV/s, LPR methods were performed on the steel reinforcement in concrete. The reinforcement was polarized at a rate of 1 mV/s within the range of potential changes from −150 mV to +50 mV regarding the corrosion potential.

The very long duration of chloride diffusion in concrete was shortened with the accelerated electromigration of chloride ions, whereas corrosion processes induced by the critical content of chloride ions in concrete took place naturally. The test specimens 1 were subjected to accelerated migration of chlorides using the electric field. Eighteen specimens grouped into six elements connected to the power source were simultaneously subjected to testing. The tests were performed in two independent test sets. Each test set was supplied with 18 V direct current 2. The specimens were placed on a big rectangular electrode (anode) made of titanium mesh 3 (coated with a thin layer of platinum) immersed in tap water at the bottom of a shallow tank 4. Plastic tanks 5 placed on the top were filled with 3% NaCl to a height of 7 cm. A round stainless-steel electrode (cathode) 6 with a diameter adjusted to the tank hole was placed on the top each specimen inside each tank. The process of chloride electrodiffusion was interrupted every 7 days to monitor the development of corrosion processes of the reinforcement by measuring corrosion potential. Electrochemical measurements were taken each time after 3 days from switching off the electricity supply to avoid polarization of the tested reinforcement (Figure 3). 

The results from polarization tests on the selected specimens C1.1; C1.2; C2.1 and C2.2 are presented in Figure 4, Figure 5 and Figure 6.

### 3.2. Material Tests—Determination of Distribution of Chloride and Hydroxide Ion Concentration at the Depth of Concrete Cover

When initiation of corrosion processes was found on the basis of interpretation of polarization measures with the LPR method, electrodiffusion was not continued and chloride profile at the depth of reinforcement cover was determined. For that purpose, “Profile Grinding Kit” by German Instruments was used to collect layers of concrete from the cover of two specimens made of each type of concrete. Crushed concrete was collected from 10 levels by 2 mm-thick layers. Later, the material from two specimens collected from the same level was mixed to average results of chloride ion concentrations in concrete (Figure 7).

By mixing crushed concrete with distilled water (1:2 ratio), ten solutions were prepared from each of the three specimens, which roughly modelled pore solution and represented averaged chemical properties of concrete tested. Concentrations of chloride ions in these solutions were measured with the multi-functional multimeter CX-701 by Elmetron with an ion-selective electrode to determine concentrations of chloride ions. Concentrations of chloride ions obtained from the chemical analysis of the tested solutions were then converted into the mean volumetric concentrations in concrete 2 mm sampling sections of total chlorides ions (${\bar{C}}_{t\;\exp}$ (mole/m^3^)) and their concentrations (C (%)) expressed for reference as a percentage of the weight of cement in concrete according to the following expressions:(1)$${\bar{C}}_{t\;\exp}=\frac{m_{t\;\exp}}{V_{c}}=\frac{2{\;c}_{\mathrm{sol}}\cdot \rho_{\mathrm{cb}}}{\rho_{w}},\;C=\frac{{\bar{C}}_{t\;\exp}}{\rho_{\mathrm{cem}}},$$
where $c_{\mathrm{sol}}$ is chloride ion concentration (kg/m^3^) determined in the tested solution, $m_{t\;\exp}$—mass (kg) of chlorides in the powdered concrete drilled from a sampling section, $V_{c}$—volume (m^3^) of the drilled concrete from sampling section in an intact state, $\rho_{\mathrm{cb}}$—bulk density (kg/m^3^) of concrete, $\rho_{w}$ = 1000 kg/m^3^—density of water,${\;\rho }_{\mathrm{cem}}$—density (kg/m^3^) of cement per a concrete unit volume. Precise details of the used experimental and calculation procedure for the determination of ${\bar{C}}_{t\;\exp}$ can be found in the article [27]. The stationery pH meter was used for simultaneous measurements conducted for all ten solutions of pore water. Calculated concentrations and pH values are shown as diagrams in Figure 4a,b.

The determined pH values were used to verify the Hausmann criterion, whose simplified version is expressed by the following expression: (2)$${\left.\frac{\left[{\mathrm{Cl}}^{-}\right]}{\left[{\mathrm{OH}}^{-}\right]}\right|}_{\mathrm{crit}.}{}_{}\leq 0.6$$

Molar concentration of chloride ions $\left[{\mathrm{Cl}}^{-}\right]$ was calculated by taking $M_{\mathrm{Cl}}=0.035453$ kg-mole as molar mass of chloride, and molar concentration of hydroxide ions $\left[{\mathrm{OH}}^{-}\right]$ was defined as pH function from the relationship $\left[{\mathrm{OH}}^{-}\right]=10^{\mathrm{pH}-14}$. Calculated concentrations and pH values are shown as diagrams in Figure 8c,d.

### 3.3. Electrochemical Chloride Extraction (ECE) with Simultaneous Control over Corrosion Processes with the Electrochemical Method

Assuming that chloride profile in relation to concrete cover depth in all of the tested specimens was similar to profiles from two tested specimens of each concrete type, the electrochemical extraction began. Such an assumption is necessary to continue testing the specimens because they are destroyed while concentration is determined. However, the averaged properties of the tested concrete were obtained by combining material from two specimens. The extraction process was performed simultaneously in six specimens 1 prepared from one concrete type and placed at a test stand similar to the one used in the electromigration process. This time, however, rebars 2 used as the cathode were connected to the negative pole 6 of 18 V DC, and the anode mesh *7* was connected to the positive pole. Tap water, into which the mesh was immersed, was used as the electrolyte. The specimens were protected with foil against drying, and water evaporating during the tests was refilled. For the group of three specimens, chlorides were extracted from each concrete type for 10 days. A week after completing the extraction process, polarization of the reinforcement was measured to evaluate effects of this process. Then, chloride profiles and pH distribution in relation to the concrete cover depth were determined within the group of two tested specimens in a way similar as described above. Extraction for the group of other two specimens of each concrete type was longer, it lasted for 11 days. Again, a week after completing the extraction process, by analogy to the above activities, polarization of the reinforcement was measured and the pore solution was tested. Figure 4 and Figure 5 illustrate results from the polarization measurements, whereas chloride profiles and distribution of pH values are shown in Figure 8. 

### 3.4. Determination of Coefficients of Migration and Extraction in Concrete

Value (D_e_ (m^2^/s))—coefficient of chloride extraction was determined, similarly to in the paper [21], on the basis of matching the diagram of chloride concentration obtained from the calculated distribution of chloride ions according to the solution of diffusion equation (where according to [21] migration coefficient was introduced into the diffusion equation) to the concentration of these ion determined during the tests and expressed with reference to the cement weight:(3)$$C_{\mathrm{cal}}=C_{0,\mathrm{cal}}\left(1-\mathrm{erf}\frac{x}{2\sqrt{D_{m,e}\;t}}\right)$$

($C_{0,\mathrm{cal}}\;\left(\%\right))$ is calculated concentration of chloride ions at the element edge with reference to the weight of cement, erf—the Gauss error function, (t(s))—duration of chloride ion migration or extraction from concrete.

To determine the most convergent computational and experimental results, the lowest s—value the mean square error, was calculated from the following expression:(4)$$s=\sqrt{\frac{\sum_{i=1}^{n}{\left[C_{\mathrm{cal}}-C\right]}^{2}}{n-1}}$$

(C(%)) is measured while measuring the concentration of chloride ions within a distance x from the element edge, $C_{\mathrm{cal}}$—chloride ion concentration within a distance x from the edge element calculated from the Equation (1), (%) chloride weight to cement weight, n—number of concrete layers, from which chloride concentration is determined. The calculated values of extraction coefficients are presented in Table 2.

## 4. Discussion of the Results Obtained

### 4.1. Results of Polarization Measurements for Reinforcement

By registering the potential changes as a function of the system response expressed as current density, a polarization curve is obtained (Figure 4 and Figure 5). The semi-logarithmic polarization curve is the basis for the graphical determination of the corrosion current density. The results of such tests are the corrosive current densities, clearly defining the corrosion rate of the reinforcement. The corrosion current $(i_{\mathrm{corr}}$ (μA)) can be calculated via polarization resistance ($R_{p}$ (kΩ)) obtained by LPR measurement according to the Stern-Geary equations [28]
(5)$$R_{p}={\left.\frac{\mathrm{dE}}{\mathrm{di}}\right|}_{i\overset{}{\to }0,\;E\overset{}{\to }E_{\mathrm{corr}}},i_{\mathrm{corr}}=\frac{b_{a}b_{c}}{2.303R_{p}\left(b_{a}+b_{c}\right)},$$
where $b_{a}$ and $b_{c}$ are constants of anodic and cathodic reactions, respectively, coefficients of rectilinear slope for segments of polarization curves—anodic $b_{a}$ and cathodic $b_{c}$. 

The corrosion current density clearly determines the corrosion intensity of steel because, according to Faraday’s law, the mass of losses ($\Delta m\;\left(\mathrm{mg}\right))$ is proportional to the flowing current ($I_{\mathrm{corr}}\;($μA/cm^2^))
(6)$$\Delta m={\mathrm{kI}}_{\mathrm{corr}}t,I_{\mathrm{corr}}=\frac{i_{\mathrm{corr}}}{A},$$
where k is electrochemical equivalent, t—time. The above relationship shows the correlation of the corrosion current density with the linear corrosion rate ($V_{r}$ (mm/year) expressed as follows:(7)$$V_{r}=0.011{\;i}_{\mathrm{corr}}$$

Corrosion rate $(V_{r}$ (mm/year) is determined from the average cross-section loss around the bar circumference, in mm, per 1 operational year of the structure. The detailed results from the analysis with the calculated densities for corrosion current are shown in Table A1 and Table A2 of the Appendix A. For an easier comparative evaluation of the obtained test results, corrosion current densities i_corr_ and E_corr_ based on values from Table A1 and Table A2, are presented in Figure 5.

The LPR tests were conducted on each test element from each measuring series. The whole period of testing produced a total of 56 polarization curves, in which exemplary shapes for four selected measuring elements are illustrated in Figure 4 and Figure 5. The specimens, for which a number of measurements were taken during the whole research process, were analysed. They were C1.1, C1.2 and C2.1, C2.2 specimens (M0: reference measurements prior to chloride migration to concrete, M1: measurement taken after 7 days of chloride migration t concrete, M2: measurement taken after another 7 days of chloride migration to concrete, E0: measurement taken after another 7 days of chloride migration to concrete and after observation of corrosion in the specimens directly prior to extraction, E1: measurement taken after 10 days of chloride extraction, E2: measurement taken after another 11 days of chloride extraction).

A similar change in distribution of polarization curves over time was observed for the specimens C1.1 and C1.2 (Figure 4). After the first reference measurement of corrosion potential, taken prior to migration $(E_{\mathrm{corr}}=205\left(C1.1\_M0\right);151\left(C1.2_{M0}\right)$ mV), other measurements taken after migration, and also after chloride extraction from concrete produced the results close to the mean value of corrosion potential ${\bar{E}}_{\mathrm{corr}}=682\left(C1.1\right)$ mV for the specimen C1.1 and ${\bar{E}}_{\mathrm{corr}}=648\left(C1.2\right)$ mV for the specimen C1.2. On the other hand, values of the measured corrosion current ranged from the reference value measured prior to migration $(i_{\mathrm{corr}}=0.21\left(C1.1\_M0\right);0.18\left(C1.2\_M0\right)$ µA) to the maximum value obtained prior to extraction $(i_{\mathrm{corr},\max}=38.94\left(C1.1\_E0\right);37.98\left(C1.2\_E0\right)$ µA). Based on the paper [29], reference values of corrosion current in the specimens C1 could be regarded as the values which signalled the unexpected corrosion, and the maximum values $i_{\mathrm{corr},\max}$ could indicate high corrosion activity.

Some similarity was also found in the behaviour of the specimens C2.1 and C2.2 made of concrete C2, but the observed trend was not the same as in the specimens made of concrete C1 (Figure 5). After the first reference measurement of corrosion potential, taken prior to migration $(E_{\mathrm{corr}}=221\left(C2.1\_M0\right);191\left(C2.2\_M0\right)$ mV), other measurements taken after migration produced the results close to the mean value of corrosion potential ${\bar{E}}_{\mathrm{corr}}=531\left(C2.1\right)$ mV for the specimen C2.1 and ${\bar{E}}_{\mathrm{corr}}=574\left(C2.2\right)$ mV for the specimen C2.2. Then, during extraction these values fluctuated around ${\bar{E}}_{\mathrm{corr}}=338\left(C2.1\right)$ mV for the specimen C2.1 and ${\bar{E}}_{\mathrm{corr}}=328\left(C2.2\right)$ mV for the specimen C2.2. On the other hand, values of the measured corrosion current ranged from the reference value $(i_{\mathrm{corr}}=0.23\left(C2.1\_M0\right);0.28\left(C2.2\_M0\right)$ µA) to the maximum value $(i_{\mathrm{corr}}=11.03\;\left(C2.2\_E0\right)$ µA). Based on the paper [29], reference values of corrosion current could be regarded as the values which signalled the unexpected corrosion, and the maximum values $i_{\mathrm{corr},\max}$ measured in the specimens made of concrete C2 indicated the moderate corrosion activity.

Figure 6a presents a comparison of results from six measurements of corrosion current density i_corr_ of the steel reinforcement in concrete from two chosen test elements made of tested concretes. Figure 6b shows a comparison of results from six measurements of corrosion potential E_corr_ of the steel reinforcement in concrete from two chosen test elements made of tested concretes. Taking into account the assumptions described in the papers [29,30], the first reference measurement taken prior to migration indicated that both corrosion potential (E_corr_ (${\bar{E}}_{\mathrm{corr}}=178\left(C1\right);206\left(C2\right))\;$< 350 mV) and corrosion current intensity (i_corr_ < 0.3 µA) suggested the passive state of all test elements. Another measurement taken after 7 days of chloride ions migration under the accelerated action of the electric field and 3 days after switching off the system indicated the onset of corrosion in all four specimens on the basis of corrosion potential values and intensity of corrosion current. However, a similar increase in average intensity of corrosion current (C1(${\bar{\Delta l}}_{\mathrm{corr}}$ = 3.91; 3.41 (C2) µA) was observed for both types of concrete. However, in the case of the first type of concrete values of corrosion, the potential increased by almost 1.5 times greater than in the second type (${\bar{\Delta E}}_{\mathrm{corr}}$ = 435 (C1); 294(C2) mV). After another 7-day charging with chloride ions, a massive increase was found for concrete C1(C1: ${\bar{\Delta l}}_{\mathrm{corr}}$ = 22.93 µA), and for concrete C2 an increase was similar as after the first week of charging and was (C2: (${\bar{\Delta l}}_{\mathrm{corr}}$ = 3.44 µA). Corrosion potential, however, increased nearly twice in concrete C2: ${\bar{\Delta E}}_{\mathrm{corr}}$ = 86 mV than in concrete C1: ${\bar{\Delta E}}_{\mathrm{corr}}$ = 48.5 mV). Then, after another 7 days of charging (and 3 days of waiting for restraining rebars) and prior to extraction, a control measurement of polarization was taken. Values obtained from this measurement suggested the developed corrosion in all of the specimens. A particularly high intensity of corrosion current i_corr_ = 38.94 µA was observed in the specimens C1.1 and C1.2 i_corr_ = 37.98 µA. Much lower values of corrosion current (C2.1(i_corr_ = 11.03 µA); C2.2(i_corr_ = 4.59 µA)) were obtained for the specimens made of concrete C2. After the first 10-day extraction, a significant drop in corrosion current intensity was observed in all four specimens. The most significant drop was found in the specimen C1.1 (Δi_corr_ = 31.53 µA) at the simultaneous drop in corrosion potential (ΔE_corr_ = 24 mV), and the smallest drop was in the specimen C2.1 (Δi_corr_ = 1.68 µA) at significantly reduced potential (ΔE_corr_ = 221 mV). After another extraction, average values of corrosion current in concrete C1 (${\bar{\Delta l}}_{\mathrm{corr}}=$ 10.87 µA) and corrosion potential (${\bar{\Delta E}}_{\mathrm{corr}}=$ 50 mV) dropped. In concrete C2, however, the average value of corrosion current slightly dropped (${\bar{\Delta l}}_{\mathrm{corr}}=$ 0.28 µA) and the average value of corrosion potential (${\bar{\Delta E}}_{\mathrm{corr}}=$ 3.5 mV) slightly increased. As it could be observed, despite of high values of corrosion current in the specimens made from concrete C1, this intensity could be, however, the average value of corrosion current slightly dropped (${\bar{\Delta l}}_{\mathrm{corr}}=$ 0.28 µA) and the average value of corrosion potential (${\bar{\Delta E}}_{\mathrm{corr}}=$ 3.5 mV) slightly increased. As it could be observed, despite the high values of corrosion current in the specimens made from concrete C1, this intensity could be “reduced” after extraction, and consequently corrosion could be inhibited due to a 10-day extraction and concentration of chloride ions on steel surface lower than the value recommended by the standards: [2,3] amounting to C_crit_ = 0.4% of cement by weight. Another week of extraction did not bring such spectacular changes in corrosion current intensity in the specimens tested, particularly in concrete C2. Although final values of corrosion potential in concrete C1(C1.1(E_corr_ = 540 mV); C1.2(E_corr_ = 752 mV) and C2 (C2.1(E_corr_ = 332 mV); C2.2(E_corr_ = 580 mV) generally showed a downward trend, when compared to maximum values measured for C1.1 (an increase by 0.1%) C1.2 (a drop by 22%), C2 (a drop by 22%); C2.2 (a drop by 40%) the majority of these results did not reach the value taken for passivated steel according to [29,30].

Although final values of corrosion current intensity in concrete C1(C1.1: i_corr_ = 3.96 µA; C1.2: i_corr_ = 1.11 µA) and C2(C2.1(i_corr_ = 3.07 µA); C2.2(i_corr_ = 3.56 µA) showed a downward trend, when compared to maximum values measured for C1.1 (a drop by 90%);) C1.2 (a drop by 23%);), C2 (a drop by 68%);); C2.2 (a drop by 33%) the majority of these results did not reach the value taken for passivated steel according to [29,30]. Similar trends were found in the papers [26,31,32,33]. 

### 4.2. Results from Material Tests on Concentration of Chloride and Hydroxide Ions in Concrete

Chloride concentration and pH were determined in pore solutions representing ten 2-mm layers. The test results obtained for two concretes are presented in Figure 8 as the distribution of chloride concentrations and pH values in the direction of the depth of reinforcement cover. As shown in Figure 9a, chloride concentration after 21 days since their electrodiffusion was at the level of ca. 0.4% [2] of cement by weight in concrete C1 near the rebar. Concentration of chloride ions along the total depth of concrete cover in concrete C2 exceeded the critical value. According to the standard criterion [2], the risk of reinforcement corrosion was probable in both cases. Measurements of reinforcement polarization in fact confirmed this assumption and indicated rather high values of corrosion current after 21 days of chloride migration to concrete, both in the specimens made of concrete C1 and C2. Howevrer, after a week of charging concrete with chloride ions corrosion current values were increasing in both concretes, which could suggest potential corrosion at concentrations of chloride ions lower than the standard concentration [2] C_crit_ = 0.4%. This deduction was confirmed by final measurements when concentration at the whole depth of concrete cover in C1 was lower than a stricter value C_crit_ = 0.1 recommended by the American standards [4,5]. The corrosion current, despite a significant drop by 90%, compared to the maximum value (the specimen C1.1) did not reach density of corrosion current assumed in the papers [29,30] as the value for steel with no risk of corrosion. When concentration of chloride ions in concrete C2 reached C_crit_ = 0.1% at the surface of reinforcing steel, corrosion current densities (C2.1(i_corr_ = 3.56 µA); C2.2(i_corr_ = 3.07 µA) were characteristic for steel with moderate corrosion activity according to the criterion described in the papers [29,30]. Very similar values were obtained for concrete C1 (C1.1(i_corr_ = 3.96 µA); C1.2(i_corr_ = 1.11 µA) at five times lower concentration of chloride ions C = 0.02% at the steel surface. 

Taking into account the additional presence of hydroxide ions, it can be observed that using the Hausmann criterion and evaluating the results from measuring corrosion current in concrete C1, the whole process of migration and extraction of chloride ions is safe and should not reach the state of corrosion risk according to this criterion. However, the measurements of corrosion did not confirm this assumption. A totally different interpretation could be made for the results obtained for the limit value of corrosion risk for the ratio [Cl^−^]/[OH^−^] ≤ 0.1.

### 4.3. Results of Chloride Extraction from Concretes

A diagram in Figure 8a presenting the distribution of chloride ion concentration after 21 days of migration, and then after 10 days and 21 days of extraction in concrete C1 leads to conclusions that a 10-day electrochemical extraction clearly reduced concentration of chloride ions at the reinforcement surface. Another 11 days of desalination only slightly reduced the concentration of chloride ions.

Concrete C2 demonstrated a similar trend. A more considerable drop in chloride ion concentration was noticed during the first phase of extraction than in the second one (Figure 8b). These observations were consistent with the observations made by other researchers [18,19], who claimed a slowdown in chlorides migration over time.

Therefore, the extraction coefficient was expected not to be the constant value during electrochemical extraction of chlorides from concrete. Moreover, the concentration at the edge of the element was observed not to be constant, but it was decreasing as the extraction progressed. Moreover, the calculated values of chloride concentrations at the element edge did not agree with the values obtained from the chemical analysis. A drop in concentration at the element edge was caused by wetting the specimen with felt, which caused rediffusion of chloride ions at the edge of the test element. The equivalent shape of concentration curves is often based on tests on in situ diffusion of chlorides when chloride ions are eluted at the element edge because of periodic rainfalls.

Based on the concentrations of chloride ions determined in the tested concretes with the method described in point 3.4, the coefficients of chloride extraction in the tested concretes were determined and their values are shown in Table 2. Figure 9 presents distribution of chloride ion concentrations determined in the tested layers of concrete n accordance with point 3.2 and calculated from the Equations (3) and (4) with the approximation method on the basis of the lowest value of mean square error. 

Based on chloride ion concentrations at the test element edge, which were determined by the approximation method of computational curves expressed by the Equations (3) and (4), a simplified theoretical linear change in chloride ion concentrations for two time intervals from 1 to 10 days and from 11 to 21 days of extraction ($t_{1}=24\;h$, $t_{2}=240\;h$, $t_{3}=504\;h)$ was taken for the computational values. Then, concentration at the element edge was calculated after the extraction time of 24; 96; 120; 144; 192; 219; 240; 288; 384; 504 h using a corresponding linear equation describing these changes (cf. Figure 10a). Similar changes in concentration at the element edge were observed for both types of concrete. A significant drop in concentration of chlorides was observed in the first-time interval, which is suggested by a wide angle of the line deviation. A drop was minor in the second stage of tests. A similar method was initially proposed by the author in the work [34] where good results were obtained for concrete with Portland cement.

Correspondingly, using previously determined values (D_e_ (m^2^/s)) of the extraction coefficient at three different times ($t_{1}=24\;h$, $t_{2}=240\;h$, $t_{1}=504\;h)$, the theoretical linear change in that coefficient was assumed for two time intervals from 1 to 10 days and from 10 to 21 days of extraction. Linear functions expressing a change in extraction coefficient are illustrated in Figure 10b. A significant drop in the extraction coefficient was observed in the first-time interval for concrete C1, which is suggested by a wide angle of the line deviation. A minor drop in the extraction coefficient was observed in both time intervals for concrete C2, which could be associated with a slowdown in chloride extraction from concrete. However, the difference in values of that coefficient was so small, that it was within the measuring error tolerance. To model extraction, extraction coefficients (D_e_ (m^2^/s)) were calculated after the following extraction times: 24; 96; 120; 144; 192; 219; 240; 288; 384; 504 h using corresponding linear equations describing these changes (cf. Figure 10b). 

Table 2 presents calculated concentrations at the element edge and the extraction coefficient, determined for 11 times of extraction from both types of concrete.

### 4.4. Predicting Duration of Extraction Using the Coefficient of Chloride Migration and Extraction

Based on the above values of concentration and the element edge, coefficients of chloride extraction and the Equation (2), the development of extraction of chloride ions over time was modelled by plotting concentration curves of these ions at the thickness of 2 cm corresponding to concrete cover thickness. Figure 8a illustrates a group of curves presenting the distribution of chloride ion concentration at the concrete cover depth of 2 cm, plotted for the following times: 24, 96, 120, 144, 192, 219, 240, 288, 384, 504 h in concrete C1. The distribution of curves indicated that after 7 days of extraction, the concentration at the steel surface was already C_(x = 20)_ = 0.1%, which should be a safe value with no corrosion risk according to the standard [2]. However, this value can present a corrosion risk according to stricter American standards [3,4]. This assumption was confirmed by results from the corrosion tests performed after 10 days of extractions. The determined densities of corrosion current C1.1(i_corr_ = 7.40 µA); C1.2(i_corr_ = 19.40 µA) were characteristic for steel with moderate corrosion activity according to the criterion described in the papers [29,30]. Another 11 days of chloride extraction did not produce such spectacular effects, but concentrations of chloride ions at the rebar surface achieved C_(x=20)_ = 0.01% from the model (and C_(x = 20)_ = 0.02% from the tests). Both of these values were lower than the boundary values specified in the standards [3,4] for compressed structures. Densities of corrosion current C1.1(i_corr_ = 3.96 µA); C1.2(i_corr_ = 1.11 µA) obtained from the tests performed after 21 days of extraction, indicated low corrosion activity according to the criterion described in the papers [29,30]. This process was effective with reference to chloride extraction from the structure to the value lower than the critical value required by the standard [2] which, unfortunately, did not guarantee safety of the structure. It should be also noted that the critical concentration C_crit_ = 0.4% at the rebar surface was achieved after 21 days of the migration process. Moreover, after the same time of extraction concentration was reduced at the rebar surface by 95%, which could suggest that extraction rate was slower than chloride ion migration to concrete within the electric field.

Figure 9b illustrates a group of curves presenting the distribution of chloride ion concentration at the concrete cover depth of 2 cm, plotted for the following times: 24, 96, 120, 144, 192, 219, 240, 288, 384, 504 h in concrete C2. The distribution of curves indicated that only after 10 days of extraction, the concentration at the steel surface was C_(x = 20)_ = 0.3% (and C_(x = 20)_ = 0.2% determined from the tests), which is a value posing a corrosion risk according to the standard [4,5]. This assumption was confirmed by results from the corrosion tests performed after 10 days of extractions. The determined densities of corrosion current C2.1(i_corr_ = 4.29 µA); C2.2(i_corr_ = 2.91 µA) were characteristic of steel with moderate corrosion activity according to the criterion described in the papers [29,30]. Another 11 days of chloride extraction did not produce such spectacular effects, but concentrations of chloride ions at the rebar surface achieved C_(x = 20)_ = 0.2% from the model (and C_(x = 20)_ = 0.1% from the tests). Both of these values are specified in the standards [4,5] as values, at which corrosion can develop. Densities of corrosion current (C2.1(i_corr_ = 3.56 µA); C2.2(i_corr_ = 3.07 µA) obtained from the tests performed after 21 days of extraction, indicated low corrosion activity according to the criterion described in the papers [29,30]. This process was effective but the critical concentration C_crit_ = 0.4% at the rebar surface was achieved after 21 days of the migration process. Moreover, after the same time of extraction, the concentration was reduced at the rebar surface by 75%, which could suggest that the extraction rate was slower than chloride ion migration to concrete within the electric field.

## 5. Conclusions and Recommendations

The performed tests demonstrated that cement CEM III/A 42.5 N-LH/HSR/NA improves protective properties of concrete against corrosion of reinforcing steel as corrosion current values were lower at higher concentrations of chloride ions. On the other hand, chloride extraction from the structure built from this type of concrete should be longer to provide as effective extraction as for concrete with cement CEM I 42.5 R. This effect can be explained by the fact that the majority of chlorides in concrete C2 were probably bounded in the cement matrix, so the real concentration of chloride ions in pore water was lower than in this type of concrete. The effect of binding chloride ions in concrete with cement CEM III/A 42.5 N-LH/HSR/NA was also confirmed in the paper [25], in which values of diffusion coefficients of chloride ions were determined, taking into account the process of binding chloride ions.

Furthermore, the determined coefficient of extraction in concrete must be considered with caution as the duration of the effective chloride extraction can be erroneously evaluated by comparing these coefficients without taking into account the distribution of concentration values at the concrete cover depth. The values of extraction coefficient in concrete C1 equal to (D_e_ = (1−6)·10^−10^ (m^2^/s)) were about 7 times higher than the values determined for concrete C2 (D_e_ = (0.3−0.9)·10^−10^ (m^2^/s)), which could suggest extraction in concrete C2 was 7 times longer than in concrete C1. Moreover, the analysed model from Figure 9b indicates that after 26 days of extraction, the concentration of chloride ions at the steel surface was C_crit_ = 0.01% and was the same as for concrete C1 after a 21-day extraction. However, it is known that too long extraction could deteriorate the mechanical properties of concrete and steel in the treated structure and increase the costs of the applied method.

The recommendations of this study:

The previously proposed method for assessing the effectiveness of electrochemical extraction of chlorides (ECE) from concrete can be successfully used to compare the effect of the cement used in concrete on the speed of the extraction process.

In another application, a rapid test on steel resistance to chloride ions can be performed using the drilled cylindrical specimen of concrete with fragments of reinforcement. This test uses the electric field to accelerate migration of chloride ions to concrete and then the densities of corrosion current are measured at the regular time intervals with the LPR method. 

A very important point here is that extraction is a longer process than migration of chloride ions into concrete. Hence, modelling the extraction process with the migration coefficient is incorrect and can lead to a too short duration of this process. As a result, this method may be ineffective.

The limitations of this study: 

The method still needs to be tested on a larger number of different concretes.

## Figures and Tables

**Figure 1 materials-16-00666-f001:**
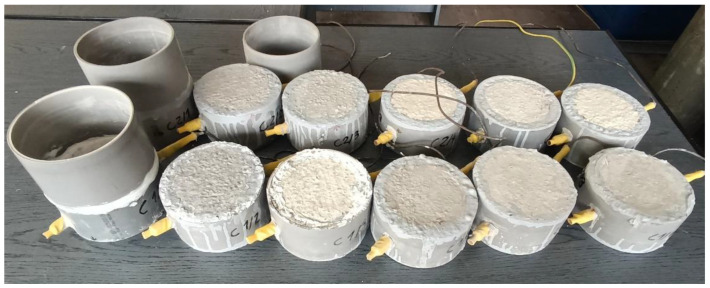
Specimens of both types of concrete prepared for testing during curing.

**Figure 2 materials-16-00666-f002:**
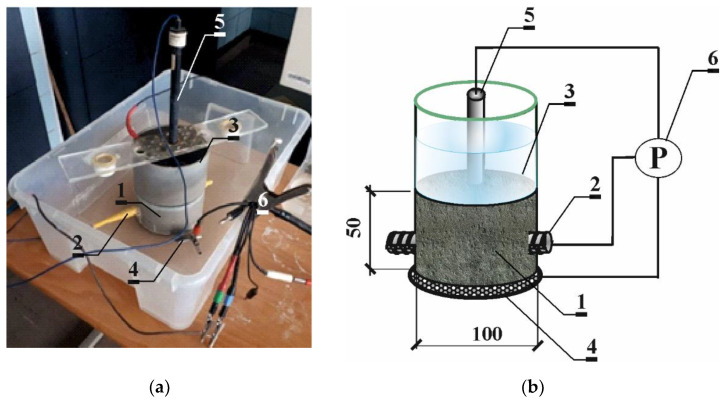
The applied test stand for polarization tests with the LPR method (**a**) view; (**b**) scheme: 1—concrete test specimen, 2—ribbed rebar ø12 mm made of steel B500S (working electrode), 3—plastic tank, 4—auxiliary electrode, 5—(Cl^−^/AgCl,Ag) electrode as the reference electrode, 6—Gamry Reference 600 potentiostat with a computer unit and Gamry software.

**Figure 3 materials-16-00666-f003:**
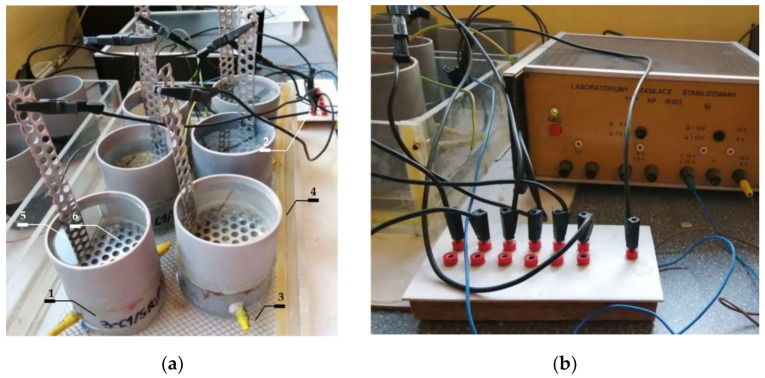
The test stand for migration of chloride ions to concrete accelerated with the electric field: (**a**) 1—concrete test specimen, 2—electric circuit of 18 V, 3—titanic anode coated with platinum, 4—tank with distilled water, 5—small plastic tanks with 3% NaCl, 6—stainless steel cathode; (**b**) stabilised laboratory feeder KP 16,103 used as source of 18 V direct current.

**Figure 4 materials-16-00666-f004:**
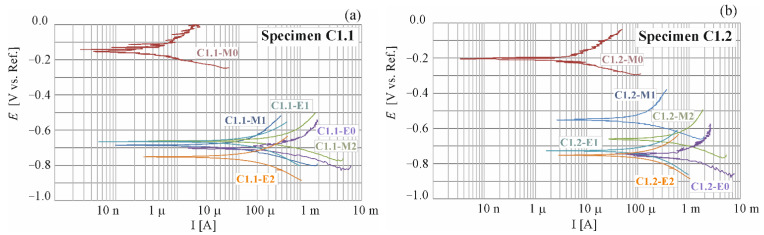
Potentiodynamic polarization curves for steel reinforcement in concrete C1 obtained for selected specimens: (**a**) C1.1 and (**b**) C1.2; M0 before chloride migration, M1 after 7 days, M2 after 14 days of chloride migration and E0 after 21 days of chloride migration and E1 after 10 days, E2 after 21 days of chloride extraction.

**Figure 5 materials-16-00666-f005:**
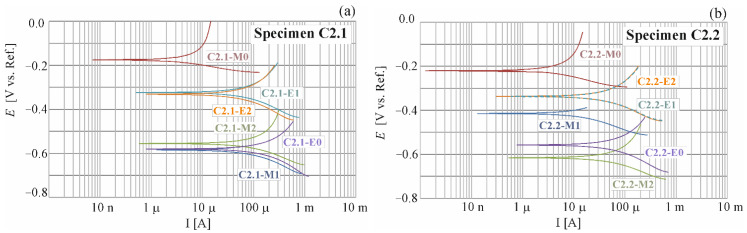
Potentiodynamic polarization curves for steel reinforcement in concrete C2 obtained for selected specimens: (**a**) C2.1 and (**b**) C2.2; M0 before chloride migration, M1 after 7 days, M2 after 14 days of chloride migration and E0 after 21 days of chloride migration and E1 after 10 days, E2 after 21 days of chloride extraction.

**Figure 6 materials-16-00666-f006:**
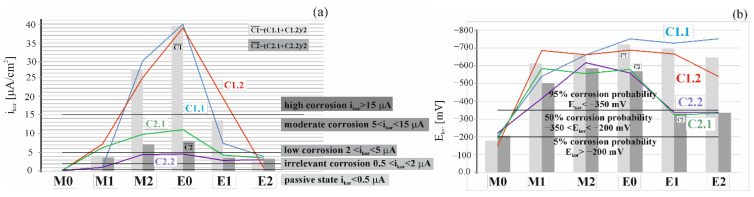
Distribution: (**a**) of corrosion current densities and (**b**) corrosion potential obtained selected specimen C1.1, C1.2, C2.1 and C2.2: M0—before chloride migration), M1—after 7 days, M2—after 14 days of migration, E0—after 21 days of migration, E1—after 10 days of chloride extraction and E2—after 21 days of extraction.

**Figure 7 materials-16-00666-f007:**
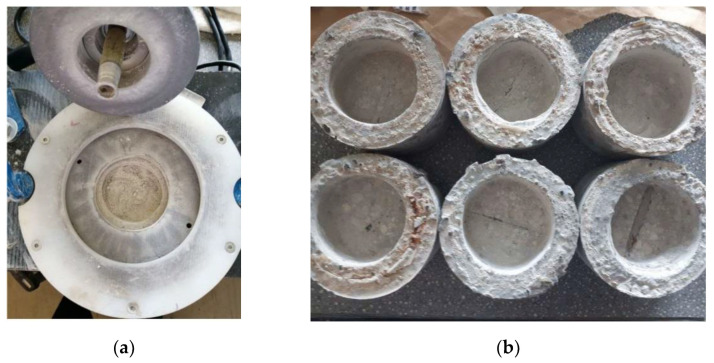
Profile Grinding Kit for concrete: (**a**) crushed concrete, (**b**) reference specimens from which concrete was collected by layers with this kit.

**Figure 8 materials-16-00666-f008:**
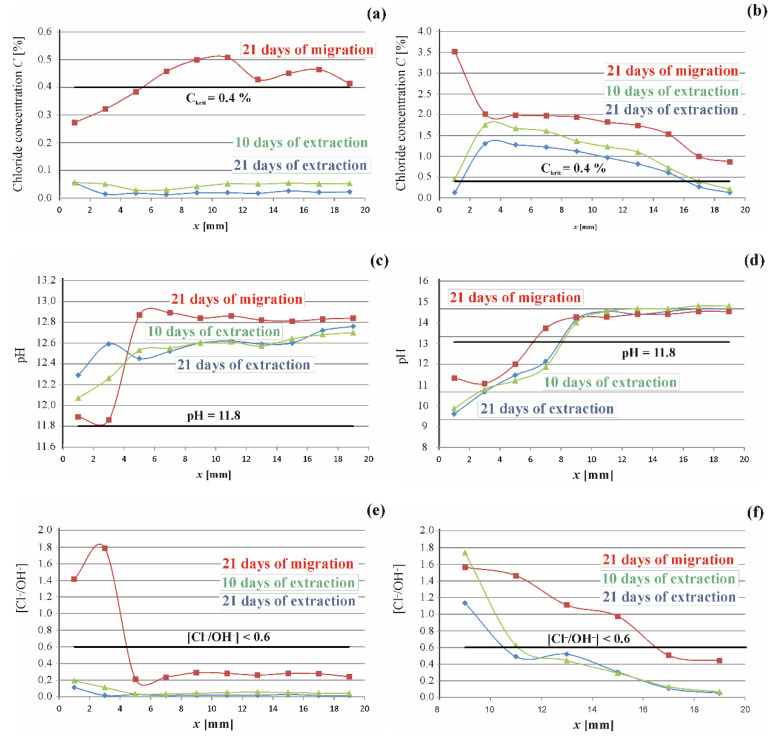
Test results obtained for reinforcement concrete C1 and C2 after 21 days of migration chloride ions, after 10 days and 21 days of extraction processes: (**a**,**b**) profiles of chloride concentrations; (**c**,**d**) distribution of pH values; and (**e**,**f**) distributions in the direction x of concrete cover thickness, values of concentration ratios of chloride and hydroxide ions—the Hausman criterion.

**Figure 9 materials-16-00666-f009:**
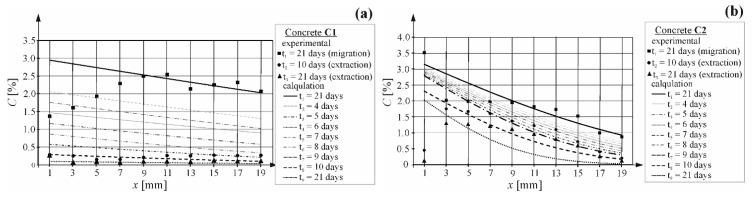
Distribution of chloride concentration in concrete obtained in the extraction process—calculated and obtained from tests: (**a**) concrete C1; (**b**) concrete C2.

**Figure 10 materials-16-00666-f010:**
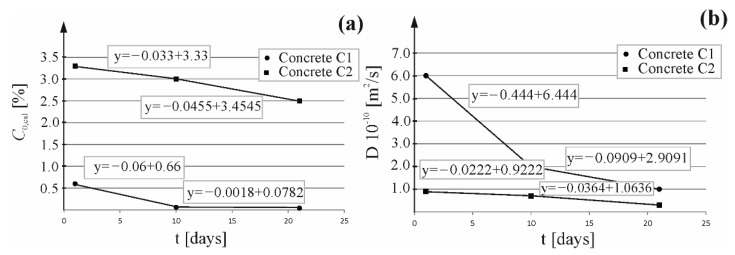
Theoretical linear change: (**a**) of the boundary chloride concentration, (**b**) of the coefficient of chloride extraction during extraction process.

**Table 1 materials-16-00666-t001:** Composition, Properties and compressive strength of concrete mixtures.

Constituent	C1	C2
	CEM I 42.5 R	CEM III/A 42.5 N-LH/HSR/NA
Cement	324	
Sand (0–2 mm)	722	
Gravel (2–8 mm)	512	
Gravel (8–16 mm)	681	
Water	162	
w/c	0.5	
Compressive strength f_cm_ MP	54.2	49.5
Volume weight $\gamma_{b}$ kg/m^3^	2271	2269

**Table 2 materials-16-00666-t002:** Numerical modeling of the extraction process.

Time of Extraction t (Hour (days))		24 (1)	96 (4)	120 (5)	144 (6)	192 (8)	216 (9)	240 (10)	288 (12)	384 (16)	504 (21)	650 (27)
initial concentration (%)	C1	0.6	0.42	0.36	0.3	0.18	0.12	0.06	0.06	0.04	0.04	-
	C2	3.3	3.2	3.17	3.13	3.07	3.03	3.00	2.91	2.73	2.50	2.27
coefficient of extraction (10^−10^ m^2^/s)	C1	5	5	4	4	3	2	2	2	1	1	-
	C2	0.9	0.83	0.81	0.79	0.74	0.72	0.70	0.63	0.48	0.3	0.12

## Data Availability

The data presented in this study are available on request from the corresponding author.

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
