# Peer review of "Evaluating the Impact of Concrete Design on the Effectiveness of the Electrochemical Chloride Extraction Process"

_materials, 2023, doi:10.3390/ma16020666_

Round 1
Reviewer 1 Report
The work submitted to the Materials journal entitled as “A new simple comparative method for evaluating the impact of concrete design on effectiveness of the electrochemical chloride extraction process” is reviewed.
In this study, presents a new simple comparative method for evaluating the impact of concrete design on effectiveness of repair by the electrochemical chloride extraction (ECE) process of reinforced concrete structures. The reviewer believes this research paper could be an interesting to material science research community and those who are interested in civil engineering.
In general, paper is well structured, and the data is well analysed and requires minor revision to be evaluated. I am suggesting the manuscript to be accepted for publication from the Materials journal however, if the authors are willing to perform minor improvements / corrections on the submitted work.
Here are the minor improvements / corrections I suggest authors to review:
- L 36 – What is “masy cementu w konstrukcjach sprężonych”
- L 42 – “In numerous papers [7], [8], [9], [10], [11], [12]” use most relevant reference and reduce the number of unnecessary reference.
- L 135 – “sand, 0÷2 mm 135 (722 kg/m3); gravel, 2÷8 mm (512 kg/m3); and gravel, 8÷16 mm (681 kg/m3)” Please follow the rules of scientific writing.
- L 151 – Include the picture of samples after preparation or during curing.
- L 156 – Please justify the reason of using ø12 mm ribbed rebars.
- L 193 – Is this testing schedule inspired from a standard or this is a novel approach in this study.
- L262 – Please justify the effects of such assumption “Assuming that chloride profile towards in relation to concrete cover depth in all the specimens tested was similar to profiles from two tested specimens of each concrete type,the electrochemical extraction began.”
- General Comments – Abstract must include numerical findings of the study.
- General Comments – Revise the keywords according to journal guidelines.
- General Comments – Article needs slight restructuring, The introduction section is quite lengthy. There are lots of unnecessary information presented. It should follow a structured literature survey.
- General comment – there is complexity on the purpose of this particular research. It should be uniquely stated what the aim is then the importance should be appreciated by those who are interested in this paper.
- General Comments – Conclusions sections should be re-arranged as Conclusion and Recommendations. In this section limitations and recommendations of this study should be listed.
- General Comments – There are some of grammatical mistakes and drawbacks in the manuscript, Please improve the English and try to present a concise expression.
- General comment – References section should be reviewed as few references are not according to the journal guidelines.
Author Response
Response to Reviewer 1 Comments
Point 1: - L 36 – What is “masy cementu w konstrukcjach sprężonych”
Response 1: It's a fragment in Polish and it means: “of cement by weight in prestressed structures.” The fragment has been corrected.
Point 2: L 42 – “In numerous papers [7], [8], [9], [10], [11], [12]” use most relevant reference and reduce the number of unnecessary reference.
Response 2: I have reduced the number of references.
Point 3: L 42 – L 135 – “sand, 0÷2 mm 135 (722 kg/m3); gravel, 2÷8 mm (512 kg/m3); and gravel, 8÷16 mm (681 kg/m3)” Please follow the rules of scientific writing.
Response 3: I have corrected the fragment according to the rules of scientific writing.
Point 4: L 42 – L 151 – Include the picture of samples after preparation or during curing.
Response 4: I have completed the picture of samples during curing prepared for testing.
Point 5: L 156 – Please justify the reason of using ø12 mm ribbed rebars.
Response 5: The most common diameter used for the main reinforcement was 12 mm and a concrete cover of 20 mm was adapted to this diameter.
Point 6: L 193 – Is this testing schedule inspired from a standard or this is a novel approach in this study.
Response 6: The test schedule was invented by the author and used for the first time for one type of concrete. In the present work, this schedule was used to compare the properties of two concretes. This is a new approach and the author is aware of the need to test the method on different concretes. It is what the author is currently working on.
Point 7: L262 – Please justify the effects of such assumption “Assuming that chloride profile towards in relation to concrete cover depth in all the specimens tested was similar to profiles from two tested specimens of each concrete type,the electrochemical extraction began.”
Response 7: Such an assumption is necessary in order to continue testing the samples, because when determining the concentration, the samples are destroyed. However, by combining the material from two samples, we obtain the averaged properties of the tested concrete.
Point 8: General Comments – Abstract must include numerical findings of the study.
Response 8: I supplemented the abstract with numerical findings of the study.
Point 9: General Comments – Revise the keywords according to journal guidelines.
Response 9: I corrected the keywords according to the journal's guidelines.
Point 10: General Comments – Article needs slight restructuring, The introduction section is quite lengthy. There are lots of unnecessary information presented. It should follow a structured literature survey.
Response 10: The introductory part has been shortened.
Point 11: General comment – there is complexity on the purpose of this particular research. It should be uniquely stated what the aim is then the importance should be appreciated by those who are interested in this paper.
Response 11: I specified the purpose of the study.
Point 12: General Comments – Conclusions sections should be rearranged as Conclusion and Recommendations. In this section limitations and recommendations of this study should be listed.
Response 12: I changed the Conclusions section to Conclusions and Recommendations and listed the limitations and recommendations of the method.
Point 13: General Comments – There are some of grammatical mistakes and drawbacks in the manuscript, Please improve the English and try to present a concise expression.
Response 13: The English language has been corrected by a native speaker.
Point 14: General comment – References section should be reviewed as few references are not according to the journal guidelines.
Response 14: Reference section has been corrected in accordance with the rules of the journal.
Thank you very much for all the reviewer's valuable comments.

Reviewer 2 Report
The paper discusses a supposedly new approach to chloride migration analysis.
The paper generally provides interesting and partly new information, but its formal and technical aspects are weak.
Examples are given:
The title is complicated and may be misleading.
The abstract contains the same sentence at the beginning and end.
The abstract is not clear enough.
The article contains a large amount of information that has already been presented, for example in citation 38, 52, 59 and 60.
The introduction of the article lacks an initial reason for evaluating chloride ions in concrete - although the article is prepared for readers with knowledge of the topic, the general focus of the journal requires an initial look at evaluating the durability of concrete. I recommend preparing a paragraph on the actual reasons for evaluating concretes, sustainability, and the rationale for similar testing and research. E.g. according to:
DOI: 10.3390/ma14247880
DOI: 10.1088/1755-1315/444/1/012021
The introductory chapter ends with a paragraph claiming that this is a new method, but it is clear from the whole article that although this is a modification of known methods, the method has been introduced by the author before.
The article contains an evaluation of two mixes and it is therefore questionable whether the topic of the article is the comparison of the two concretes or the method itself - as such the article misses its main purpose.
Even the final conclusions are contradictory - the first part shows that the method you presented creates error and is prone to evaluation problems. The second part of the conclusion on the evaluation of CEM III/A 42.5 N-LH/HSR/NA concrete itself contains the information that the concrete has already been analysed, so it is questionable whether the article adds anything new when the information is already available elsewhere.
Author Response
Response to Reviewer 2 Comments
Point 1: - The title is complicated and may be misleading.
Response 1: The title of the article has been simplified.
Point 2: L 42 – The abstract contains the same sentence at the beginning and end.
Response 2: The abstract has been edited again.
Point 3: The abstract is not clear enough..
Response 3: The abstract has been edited again.
Point 4: L 42 – L 151 – The article contains a large amount of information that has already been presented, for example in citation 38, 52, 59 and 60.
Response 4: The information in this section has been abbreviated.
Point 5: L 156 – The introduction of the article lacks an initial reason for evaluating chloride ions in concrete - although the article is prepared for readers with knowledge of the topic, the general focus of the journal requires an initial look at evaluating the durability of concrete. I recommend preparing a paragraph on the actual reasons for evaluating concretes, sustainability, and the rationale for similar testing and research. E.g. according to:
DOI: 10.3390/ma14247880
DOI: 10.1088/1755-1315/444/1/012021.
Response 5: The introduction was supplemented according to the reviewer's comments.
Point 6: The introductory chapter ends with a paragraph claiming that this is a new method, but it is clear from the whole article that although this is a modification of known methods, the method has been introduced by the author before.
Response 6: The method was introduced for one concrete. In this article, it was tested in a different concrete and used to compare both concretes at the same time. In the context of comparing concretes, it is a new method.
Point 7: L 42 – The article contains an evaluation of two mixes and it is therefore questionable whether the topic of the article is the comparison of the two concretes or the method itself - as such the article misses its main purpose.
Response 7: The article has two goals:
- the first is to test the method on the second concrete (it will be necessary to test on many different concretes)
- the second goal is to compare the effectiveness of Electrochemical Chloride Extraction (ECE) in both these concretes using this method
Point 8: Even the final conclusions are contradictory - the first part shows that the method you presented creates error and is prone to evaluation problems. The second part of the conclusion on the evaluation of CEM III/A 42.5 N-LH/HSR/NA concrete itself contains the information that the concrete has already been analysed, so it is questionable whether the article adds anything new when the information is already available elsewhere.
Response 8: The ECE method itself is susceptible to evaluation problems. I am trying to eliminate these problems in my scheme.
Z drugiej strony z cementem CEM III nadal sprawia problemy i nie jest jasne, czy jego zastosowanie w budownictwie jest korzystne, szczególnie podczas dÅ‚ugotrwaÅ‚ej eksploatacji w kontekÅ›cie trwaÅ‚oÅ›ci konstrukcji.
Bardzo dziękuję za wszystkie cenne uwagi recenzenta.

Reviewer 3 Report
This paper presents a method for evaluating the impact of concrete design on effectiveness of repair by the electrochemical chloride extraction (ECE) process of reinforced concrete structures.
1But some papers [13], [14] described that pH increase above 12.6 resulted in a significant drop of fixed chloride. It means that an increase in corrosion risk is associated with an increase in chloride ion concentration despite the inhibiting action of hydroxyl ion,please make sure it.
2Powyższa zależność pozwala na powiÄ…zanie gÄ™stoÅ›ci prÄ…du korozyjnego z liniowÄ… szybkoÅ›ciÄ… korozji (??(mm/year)okreÅ›lonej równaniem?
3 Compared with corrosion potential measurement, where does A new simple comparative method appear.
Author Response
Response to Reviewer 3 Comments
Point 1: - -But some papers [13], [14] described that pH increase above 12.6 resulted in a significant drop of fixed chloride. It means that an increase in corrosion risk is associated with an increase in chloride ion concentration despite the inhibiting action of hydroxyl ion,please make sure it.
Response 1: It is difficult to control the level of concentration of OH ions in concrete, which can change with the change of concrete moisture. Therefore, in the presence of chloride ions, a high level of pH value does not guarantee the safety and passivity of the steel.
Point 2: L 42 – - The above relationship shows the correlation of the corrosion current density with the linear corrosion rate (V_r (mm/year) expressed as follows?
Response 2: Based on Faraday's law, a linear relationship is assumed between the corrosion rate and the corrosion current.
Point 3: Compared with corrosion potential measurement, where does A new simple comparative method appear..
Response 3: Measuring the corrosion potential alone is insufficient to determine the rate of corrosion. Only more accurate LPR or EIS measurements allow to quantify the corrosion rate, which in the case of treatment of an object with the ECE method is of great importance because these processes can both be overestimated or underestimated.

Reviewer 4 Report
The article, "A new simple comparative method for evaluating the impact of 2 concrete design on effectiveness of the electrochemical chloride 3 extraction process," shows an in-depth study on corrosion and chloride extraction from concretes prepared with two different types of cement. Although the article presents interesting data, more revisions are recommended. The critical points below:
-English in various parts of the article is poor and often difficult to read, plus parts written in Polish were still found at line 36 and between lines 321-323.
- The quality of the figures is very bad and does not allow the graphs to be read correctly, also in some scales the units seem to be wrong according to the scale entered.
-Figure and table captions are too long and often tend to be confusing; I suggest explaining the figures and tables in the text rather than in the captions.
- It was stated that a type I 42.5 cement was used, this is already an anti-corrosive cement especially from chlorides, however, the author in the text did not take this into account, in fact in the conclusion part stated that it is more difficult to extract chlorides from such a cement. I would therefore invite further analysis of why it is more complicated than the other cement used.
Author Response
Response to Reviewer 4 Comments
Point 1: - - English in various parts of the article is poor and often difficult to read, plus parts written in Polish were still found at line 36 and between lines 321-323.
Response 1: The text has been corrected by a native speaker.
Point 2: L 42 – - The quality of the figures is very bad and does not allow the graphs to be read correctly, also in some scales the units seem to be wrong according to the scale entered.
Response 2: The readability of the figures and individual units has been improved.
Point 3: Figure and table captions are too long and often tend to be confusing; I suggest explaining the figures and tables in the text rather than in the captions.
Response 3: Descriptions of tables and figures have been shortened.
Point 4: L 42 – L 151 – It was stated that a type I 42.5 cement was used, this is already an anti-corrosive cement especially from chlorides, however, the author in the text did not take this into account, in fact in the conclusion part stated that it is more difficult to extract chlorides from such a cement. I would therefore invite further analysis of why it is more complicated than the other cement used.
Response 4: In fact, cement type I 42.5 is recommended as more resistant to chlorides, especially in bridge structures. However, there are many publications suggesting better resistance of type III/A 42,5 composite cement. This phenomenon may be related to the greater ability to bind chlorides by multi-component cement, which on the one hand protects the structure for longer. However, after the ions penetrate to a dangerous depth, it becomes more difficult to treat the structure.

Round 2
Reviewer 2 Report
Thank you for the edits. I can't comment further.
Author Response
Thank you for all your valuable comments.
Reviewer 3 Report
1 The reinforcement was polarized at a rate of 1 mV/s within the range of potential changes from −150 mV to +50 mV regarding the corrosion potential. When initiation of corrosion processes was found on the basis of interpretation of polarization measures with the LPR method.Please be careful what LPR means.
2 "This paper presents a new simple comparative method for evaluating the impact of 86 concrete design on the effectiveness of electrochemical extraction chlorides from concrete."Still here
Author Response
Point 1: The reinforcement was polarized at a rate of 1 mV/s within the range of potential changes from −150 mV to +50 mV regarding the corrosion potential. When initiation of corrosion processes was found on the basis of interpretation of polarization measures with the LPR method. Please be careful what LPR means.
Response 1: Linear polarization resistance (LPR) is a quick, nondestructive testing technique commonly used in material corrosion studies to gain corrosion rate data. For this method the material is polarized, typically on the order of ±10mV, relative to its Open Circuit (OC) potential—the potential measured when no [net] current is flowing. As the potential of the material (working electrode) is changed, a current will be induced to flow between the working and counter electrodes, and the material’s resistance to polarization can be found by taking the slope of the potential versus current curve. This resistance can then be used to find the corrosion rate of the material using the Stern-Geary equation.
The polarity of the potential can be a source of confusion. In electrochemical corrosion measurement, the equilibrium potential assumed by the metal in the absence of electrical connections to the metal is called the open-circuit potential, Eoc. We use the term corrosion potential, Ecorr, for the potential in an electrochemical experiment at which no current flows, as determined by a numerical fit of current-versus-potential data. In an ideal case, the values for Eoc and Ecorr are identical. One reason the two voltages may differ is that changes have occurred to the electrode surface during the scan. Regardless of whether potentials are versus Eref or versus Eoc, one sign convention is used. The more positive a potential, the more anodic it is. More anodic potentials accelerate oxidation at the Working Electrode. Conversely, a negative potential accelerates reduction at the Working Electrode.
I am aware of some disadvantages of the LPR method for measuring corrosion in a material such as concrete. Thanks to the use of the Gamry device and the company's software, I assisted in the interpretation of the test results and I hope that all the inconveniences have been reduced. The generated current was registered and the ohmic drop was compensated.
Point 2: "This paper presents a new simple comparative method for evaluating the impact of concrete design on the effectiveness of electrochemical extraction chlorides from concrete." Still here
Response 2: Sorry, the part was overlooked. Already has been removed.

Reviewer 4 Report
The author has responded to all critical issues in the article, so it can be accepted for publication.
Author Response

(The authors gave the same response as above.)
